# Unraveling the Phylogenomic Relationships of the Most Diverse African Palm Genus *Raphia* (Calamoideae, Arecaceae)

**DOI:** 10.3390/plants9040549

**Published:** 2020-04-23

**Authors:** Andrew J. Helmstetter, Suzanne Mogue Kamga, Kevin Bethune, Thea Lautenschläger, Alexander Zizka, Christine D. Bacon, Jan J. Wieringa, Fred Stauffer, Alexandre Antonelli, Bonaventure Sonké, Thomas L. P. Couvreur

**Affiliations:** 1IRD, DIADE, University Montpellier, 34394 Montpellier, France; andrew.j.helmstetter@gmail.com (A.J.H.); kevin.bethune@hotmail.com (K.B.); 2Laboratoire de Botanique systématique et d’Ecologie, Department of Biological Sciences, University of Yaoundé I, Higher Teacher Training College, Yaoundé B.P. 047, Cameroon; mogueblue@yahoo.com (S.M.K.); bsonke_1999@yahoo.com (B.S.); 3Institute of Botany, Department of Biology, Faculty of Science, Technische Universität Dresden, 01062 Dresden, Germany; thea.lautenschlaeger@tu-dresden.de; 4German Center for Integrative Biodiversity Research (iDiv) Halle-Leipzig-Jena, 04103 Leipzig, Germany; alexander.zizka@idiv.de; 5Department of Biological and Environmental Sciences and Gothenburg Global Biodiversity Centre, University of Gothenburg, 405 30 Gothenburg, Sweden; christine.bacon@bioenv.gu.se (C.D.B.); a.antonelli@kew.org (A.A.); 6Gothenburg Global Biodiversity Centre, Box 461, SE 40530 Goteborg, Sweden; 7Naturalis Biodiversity Center, Darwinweg 2, 2333 CR Leiden, The Netherlands; jan.wieringa@naturalis.nl; 8Department of Botany and Plant Biology, Conservatory and Botanical Garden of the City of Geneva, University of Geneva, 1205 Geneva, Switzerland; Fred.Stauffer@ville-ge.ch; 9Royal Botanic Gardens, Kew, Richmond, Surrey TW9 3AE, UK

**Keywords:** Africa, exons, Madagascar, rain forests, phylogenomics, *Raphia*, sequence capture

## Abstract

Palms are conspicuous floristic elements across the tropics. In continental Africa, even though there are less than 70 documented species, they are omnipresent across the tropical landscape. The genus *Raphia* has 20 accepted species in Africa and one species endemic to the Neotropics. It is the most economically important genus of African palms with most of its species producing food and construction material. *Raphia* is divided into five sections based on inflorescence morphology. Nevertheless, the taxonomy of *Raphia* is problematic with no intra-generic phylogenetic study available. We present a phylogenetic study of the genus using a targeted exon capture approach sequencing of 56 individuals representing 18 out of the 21 species. Our results recovered five well supported clades within the genus. Three sections correspond to those based on inflorescence morphology. *R. regalis* is strongly supported as sister to all other *Raphia* species and is placed into a newly described section: Erectae. Overall, morphological based identifications agreed well with our phylogenetic analyses, with 12 species recovered as monophyletic based on our sampling. Species delimitation analyses recovered 17 or 23 species depending on the confidence level used. Species delimitation is especially problematic in the Raphiate and Temulentae sections. In addition, our clustering analysis using SNP data suggested that individual clusters matched geographic distribution. The Neotropical species *R. taedigera* is supported as a distinct species, rejecting the hypothesis of a recent introduction into South America. Our analyses support the hypothesis that the *Raphia* individuals from Madagascar are potentially a distinct species different from the widely distributed *R. farinifera*. In conclusion, our results support the infra generic classification of *Raphia* based on inflorescence morphology, which is shown to be phylogenetically useful. Classification and species delimitation within sections remains problematic even with our phylogenomic approach. Certain widely distributed species could potentially contain cryptic species. More in-depth studies should be undertaken using morphometrics, increased sampling, and more variable markers. Our study provides a robust phylogenomic framework that enables further investigation on the biogeographic history, morphological evolution, and other eco-evolutionary aspects of this charismatic, socially, and economically important palm genus.

## 1. Introduction

Palms are iconic floristic elements across the tropics both in terms of diversity and the natural resources they provide, playing important roles for the welfare of rural and urban people at equatorial latitudes. Worldwide, there are an estimated 2500 palm species [1], mainly occurring in tropical rain forests. Africa, however, harbors less than 70 species (excluding Madagascar) [2,3], a pattern that contrasts strongly with the Neotropics or South East Asia, which contain 800 and 1200 species, respectively [1,4,5]. Despite this low diversity, palms are omnipresent across the African landscape, particularly in the tropical rain forests of the continent [2,6].

Among African palms, the genus *Raphia* (subfamily Calamoideae, tribe Raphiaeae) is the most species rich, with 21 species described to date [2,7]. Of these, one, *R. taedigera* Mart., is endemic to the Neotropics, with a disjunct distribution in Brazil and central America. The presence of this species in the Neotropics was suggested as either pre-Colombian and natural (biogeographic long distance dispersal/vicariance [8,9]) or as recently naturalized by Africans during the slave trade some 400 years ago [6,10,11]. *Raphia* species mainly occur in tropical rain forests, most often in swampy or periodically inundated areas where they can dominate the vegetation, producing dense monospecific stands (known as “Raphiales” in French). A few species, however, have adapted to drier conditions restricted to river systems in the Sahel or southern Africa.

*Raphia* is one of the most economically important genus of African palms across tropical African communities. One recent study documented over 100 different uses across the genus, with the most important ones being extraction of palm wine, grubs and construction material [12,13]. Exploitation of its species in the wild also represents an important source of income for populations across tropical Africa, especially for low-income households [12,14,15]. In addition, *Raphia* species play vital ecological roles in wet land ecosystems [16] where they dominate the landscape, such as in peatlands of the Congo Basin where they are highly abundant [17]. *Raphia* dominated swamps are also important ecosystems for the protection for critically endangered animals, such as lowland gorillas because such areas are hard to access or to bring into cultivation (e.g., [18]).

*Raphia* species are massive palms with very long pinnate leaves. One species holds the record for the longest measured leaf in angiosperms, reaching up to 26 meters (*R. regalis*). The stipe is generally above-ground and is solitary or clustered, while three species have very short (*R. palma-pinus* (Gaertn.) Hutch. and *R. vinifera* P. Beauv.) or subterranean (acaulescent) (*R. regalis* Becc.) stipes. When present, the stipe can be covered by old leaf sheath remains or a dense network of fibers (decomposed leaf sheaths), which can be curly or straight, an important character to identify species (e.g., [19,20]). *Raphia* species are monoecious, with male and female flowers on the same individual and are hapaxanthic, meaning that individual stipes die after a single flowering event [1]. The inflorescence structure is relatively simple and branched to two orders [1]. The first and second order branches, or rachillae, are referred to as the “partial inflorescence“ [21]. The shape and overall morphology of these partial inflorescences are one of the most important taxonomic characters for species identification and were used to define the different sections within the genus [20,21].

Despite its importance, *Raphia* remains one of the least understood palm genera in terms of taxonomy and phylogenetic relationships [1,20]. This is mainly due to their massive size, making them difficult to collect for non-specialists, which leads to few informative herbarium specimens or specimens that are incomplete or fragmentary and hence uninformative for taxonomy. Several attempts have been undertaken to tackle the taxonomy of the genus, beginning in the early 1900s with the first complete monograph of the genus [22]. This was followed by more regional attempts through the last century [23,24]. The last major revision of the genus was undertaken by Otedoh [21], who placed species into five different sections based on the structure of the partial inflorescence: Moniliformes (including the subsection Erectae), Temulentae, Raphiate, Flabellatae, and Obclavatae.

The six species within the Moniliformes section are characterized by thin and easily breakable rachillae when fresh (Figure 1B). Otedoh [21] also created a subsection, Erectae, where he placed two species in which the inflorescences are defined as erect (*R. australis* Oberm. and Strey, *R. regalis*) (Figure 1G,O). The Temulentae section has robust and tightly appressed rachillae. The partial inflorescences are racquet-shaped with the apical second order rachillae shorter than the basal ones (Figure 1E). This section contains three (possibly four) species, including one of the most widespread and important species *R. hookeri* G. Mann & H. Wendl. With seven species, the Raphiate section is the most complex group of the genus. Several species in this section are only known from a few collections or just the type. This section is characterized by species having second order rachillae that are robust (thick) but loosely disposed between them (Figure 1D). The inflorescence within this section can be erect, semi-erect, or drooping (Figure 1I). The Flabellatae section contains two species with very characteristic partial inflorescence structures. The second order rachillae are tightly packed in a single plane being racket-shaped in appearance (Figure 1F). The inflorescence also has very conspicuous bracts that cover completely or partially the partial inflorescences (Figure 1O). Finally, the Obclavatae section contains one species (*R. sudanica* A. Chev.) with distinct club-shaped and compact partial inflorescences with large bracts covering too (Figure 1C).

To date, no in depth morphological or molecular phylogenetic study of *Raphia* has been undertaken. The most recent phylogenetic analysis of the Calamoideae subfamily included a single species and individual of *Raphia*, namely *R. farinifera* (Gaertn.) Hylander sequenced for the ribosomal region ITS and the plastid region rps16 [25]. The main objective of this study was to generate a densely sampled phylogenetic tree of the genus and test the validity of the taxonomic sections of Otedoh [21]. In particular, we tested if the partial inflorescence structure has a phylogenetic signal and is useful for *Raphia* species classification. In addition, by sampling several individuals per morphologically identified species, we also tested species delimitation and monophyly. In order to achieve these objectives, we sequenced more than 150 palm specific nuclear markers across 56 *Raphia* accessions. We used a species delimitation approach to define species limits and generated SNP data to study at fine-scale genetic relationships in identified species complexes.

## 2. Results

### 2.1. DNA Sequencing

We sequenced 56 individuals representing 18 species or 87.5% of the species diversity within the genus. A total of 15.4 million reads were generated and mapped to the reference exons belonging to 176 genes of the Heyduk et al. [26] baiting kit. Across all *Raphia* and outgroup individuals, the average coverage depth was 139.6x. We identified 102 genes for which 75% of the exon length was recovered in at least 25% of individuals. Twenty loci were flagged by Hybpiper as paralogs because multiple assembled contigs matched a single reference locus. Of these 20, those that occurred in the 75/25 set were removed, resulting in a final dataset of 85 supercontigs equaling 162 kb of sequence data. Our SNP calling approach applied filters on mapping quality (>40%) depth (>25), quality by depth (>2), minimum depth across individuals (>10) minor allele frequency (>0.01), and we excluded monomorphic sites. This ultimately yielded 915 and 1627 high-quality, biallelic SNPs for the *R. hookeri* and *R. zamiana* species complexes, respectively (see below). The fastq (R1 and R2) sequences for all individuals are available in Genbank’s Sequence Read Archive (SRA) under Bioproject number PRJNA615688 (http://www.ncbi.nlm.nih.gov/bioproject/615688).

### 2.2. Evolutionary History of Raphia

We generated two phylogenetic hypotheses for *Raphia* using two distinct methods. The first analysis was conducted based on a gene-tree coalescent approach using ASTRAL, while the second inferred phylogenetic relationships were based on a concatenated approach using IQ-TREE.

Support varied throughout the *Raphia* ASTRAL tree—About 50% of branches had a local posterior probability (LPP) above 75% (Figure 1A). Major clades were well supported (LPP > 80%) while relationships towards the tips of the tree generally had lower support. The final normalized quartet score, the proportion of quartet trees that agree with the species tree, was 65%, indicating that there is gene tree conflict in the genus.

The IQ-TREE concatenated approach (see Figure A1 in Appendix B) had increased bootstrap support compared to ASTRAL. More than 88% of branches had bootstrap support greater than 75%. The best partitioning scheme put the 85 loci into 20 different partitions. Major clades were again well-supported in this tree (bootstrap > 80%).

Our phylogenetic analyses recovered five well supported clades. Three clades matched the sections as defined by Otedoh [21]. *Raphia regalis* was always inferred with strong support as sister to the rest of the genus independent of the inference method (Figure 1 and Figure A1). When comparing the two phylogenetic approaches, we identified a topological difference in the phylogenetic placement of the section Temulentae, the species *R. matombe*, and the Moniliformes and Flabellatae sections (Figure 2). In the IQ-TREE analyses, we recovered weak support for the Temulentae as sister to all *Raphia* (except *R. regalis*) (Figure 2A) while the ASTRAL analysis indicates with higher support that Temulentae is sister to a clade containing *R. matombe*, Moniliformes and Flabellatae (Figure 2B).

The relationships between species in the Raphiate section are weakly to moderately supported in both analyses (Figure 1 and Figure A1). Nevertheless, we do recover monophyletic groups in some species consistent with prior morphological identifications. This is the case for individuals of *R. laurentii* and *R. monbuttorum*, which despite low support are monophyletic. Furthermore, both of these species are recovered as sister, with moderate to high support. However, our species delimitation analysis suggested that individuals identified under both species are conspecific (Figure 1A). Support is generally higher in the ASTRAL tree, even when taking into account different gene histories, so we suggest that the ASTRAL tree represents a more accurate reconstruction of the phylogeny of *Raphia* (Figure 2B) so we will principally refer to the relationships in this tree from now on.

### 2.3. Species Delimitation

Our species delimitation approach yielded between 17 (α = 0.005) and 23 (α = 0.01) species (Figure 1). Higher values of α split a clade of closely related individuals (marked with a star in Figure 1), predominantly belonging to *R. hookeri*, into seven different species. Generally, our species delimitation results corresponded well with our field identifications and using available floras (e.g., [19,20]). In some cases, we found that SODA split individuals belonging a priori to a single species into multiple species—for example, *R. farinifera* and *R. sudanica* (Figure 1). Conversely, individuals assigned to different species such as *R. laurentii* and *R. monbuttorum* were classified as the same species according to SODA delimitation independent of α values. In general, the support among different species as delimited by SODA was high (Figure 1).

### 2.4. Fine Scale Structure in Two Species-Complexes

To further explore genetic structure among our two main species complex, namely the “zamiana“ and “hookeri“ complexes (marked with a black square and a black star in Figure 1, respectively), we used SNPs extracted from the sequence data to look at the variation among individuals. The “hookeri“ complex showed little evidence of clustering, with most individuals evenly spread out on the first two principal component (PCA) axes (Figure 3a). We observed two major groups of >8 individuals in the “zamiana“ complex along PCA1 (Figure 3b), separating all of the *R. laurentii* and *R. monbuttorum* from the rest of the individuals. The first two PCAs in both analyses explained 7–10% of the variance in the dataset. This is a still less than a quarter of the total variance in each case, but this may be expected from our relatively complex dataset of almost 100 genes with independent histories. In general, our SNP data support SODA species delimitation as the assigned species grouped together along one or both of the first two PCAs in most cases (Figure 3a,b). Finally, our SNP data revealed that individuals within the “hookeri“ complex clustered into four major groups (Figure 3a,c): the single individual from Togo; individuals from western Cameroon; individuals from East Cameroon and individuals from Gabon.

The plotting of these complexes on maps of the sampling region revealed that the delimited species clustered geographically (Figure 3c,d). In the “hookeri“ complex, the *R. sese* individuals were sampled at a great distance from each other and *R. gabonica* falls in the middle of the *R. hookeri* distribution range. Likewise, in the “zamiana“ complex, *R. laurentii* and *R. monbuttorum* are widespread, overlapping with other taxa. Many of the delimited species co-occur or are adjacent to one another in Cameroon.

## 3. Discussion

### 3.1. Synthesizing Morphology and Molecules: The Sections of Otedoh Reevaluated

Our phylogenomic analyses of *Raphia* provide a novel and overall well supported phylogenetic framework for this important African genus (Figure 1A). Although some of the morphology-based sections of Otedoh [21] were recovered, we also recovered some topological differences (Figure 1A).

The Moniliformes and Flabellatae are not recovered as monophyletic. The Moniliformes are split into two clades, while the two Flabellatae species (*Raphia farinifera*, *R. vinifera*) are not recovered as sister and are nested within the Moniliformes section (Figure 1). In addition, the phylogenetic placement of the Moniliformes species *R. matombe* from the Democratic Republic of the Congo and Angola (including Cabinda) was different between the two types of analyses (Figure 2).

In all analyses, the acaulescent central African species *Raphia regalis* is recovered with strong support as sister to the rest of the genus (Figure 1A and Figure A1). This species, together with *R. australis*, were placed within the subsection “erectae” within the Moniliformes section [20,21] because the inflorescences were suggested to be “erect”, in contrast to the rest of the *Raphia* species whose inflorescences are hanging or semi-erect (except for *R. palma-pinus* which also has an erect inflorescence). Our results do not support this classification, as *R. australis* is recovered as sister to *R. farinifera* (of the Flabelattae section, Figure 1) and phylogenetically divergent from *R. regalis*. A closer observation in the field showed that only the inflorescences of *R. australis* are truly erect (Figure 1N). In contrast, the inflorescences of *R. regalis* appear erect but are in fact “supported” by the large leaves and not truly erect (Figure 1G).

The close relationships between the Moniliformes and Flabellatae section are not surprising. The inflorescences, although different in some aspects such as the clearly racket-shaped partial inflorescences of the Flabellatae section, show certain similarities not encountered in other *Raphia* species. Both have thin rachillae and the partial inflorescences are subtended by large showy bracts at least in the younger stages of development. These morphological similarities thus support the close phylogenetic relationships recovered here between these two sections.

The Obclavate section, composed of the sole species *R. sudanica*, is recovered with strong or moderate support as sister to the Raphiate section. This species presents a unique inflorescence structure within the genus that is reduced and compressed into a cylindrical shape (Figure 1C), with large bracts covering the partial inflorescences almost completely [20,21,27]. In addition, and in contrast to most species, *R. sudanica* thrives within the drier regions of the Sahel along small stream courses. These distinctive characters and its phylogenetic position support it being placed in a section of its own, confirming the classification of Otedoh [21].

Finally, the two remaining sections, Raphiate and Temulentae, are recovered as monophyletic, although with varying levels of support from strong to moderate (Figure 1A and Figure A1). This also confirms the classification of Otedoh [21] and the usefulness of partial inflorescence shapes in the classification of *Raphia* species.

Our results, however, suggest that certain sections erected by Otedoh [21] are not monophyletic and need to be re-evaluated. Differences in phylogenetic relationships between the concatenated and coalescent approaches have been increasingly reported in this genomic era [28,29,30]. Our results were similar to those in Couvreur et al. [19] where higher bootstrap support was obtained when using the concatenation approach, despite the coalescent approach highlighting considerable gene tree conflict. Here, we favor the phylogenetic hypothesis recovered when using the coalescent approach (Figure 1) because these methods allow gene history to be taken into account [31] and provide an arguably more realistic reconstruction of phylogenetic relationships when using a large number of independently evolving nuclear markers as used here. Our analyses suggest that we can retain five sections, only slightly different than those initially defined by Otedoh [21]. Three sections have been reconstructed in the phylogeny: Obclavatae (with its only species *R. sudanica*), Raphiate, and Temulentae. The latter two sections are internally complicated, and more discussion about the phylogenetic relationship within sections is provided below. The main problem thus comes from the Moniliformes and Flabellatae sections, which are not monophyletic. *Raphia regalis* should be placed in a section of its own, linked to its unique morphology being an acaulescent species with large inflorescences subtended by large leaves (Figure 1G). Finally, the last section should regroup all the other species from both the Moniliformes and Flabellatae.

We thus recognize five main sections within *Raphia* based on the phylogenetic results presented here. We then discuss these results in more detail below.

1. Section Erectae (Otedoh) Couvreur, Mogue, and Sonké, **sect. nov.**. Type species: *R. regalis* Becc.

Diagnose: Acaulescent palms with less than ten leaves. Inflorescences erect amongst leaves, rachillae thin, and brittle. Although the inflorescences are not truly erect (see above), we prefer to conserve the name “Erectae” for this new section. Contains to date one species: *R. regalis*.

2. Section Moniliformes Otedoh. Type species: *R. textilis* Welw. This section also includes the species formally placed in section Flabellatae by Otedoh [21]. Because the name Moniliformes was published before Flabellatae (page 148 versus 163), we retained the former name here. A section with seven recognized species: *R. australis*, *R. farinifera*, *R. gentiliana* De Wild., *R. matombe* De Wild., *R. ruwenzorica* Otedoh, *R. textilis* Welw. and *R. vinifera* P.Beauv. (sensus Kamga Mogue et al. [32]).

3. Section Temulentae Otedoh. Type species: *R. hookeri* Mann and Wendl. This section remains the same as defined by Otedoh [21]. We also include the newly described species *R. gabonica* [7] in this section.

A section with four recognized species: *R. gabonica* Mogue, Sonké, Couvreur, *R. hookeri* Mann and Wendl., *R. sese* De Wild. and *R. rostrata* Burret.

4. Section Raphiate Otedoh. Type species: *R. palma-pinus* (Gaertn.) Hutch.. Otedoh [21] had an erroneous vision of *R. vinifera* (the type of this section) which does not belong to the Raphiate section (see [32] for details). We thus choose a new type species being the second oldest name within this section.

A section with eight recognized species: *R. africana* Otedoh, *R. laurentii* De Wild., *R. longiflora* Mann and Wendl., *R. mannii* Becc., *R. monbuttorum* Drude, *R. palma-pinus* (Gaertn.) Hutch., *R. taedigera* Mart. and *R. zamiana* Mogue, Sonké, Couvreur.

5. Section Obclavatae Otedoh. Type species: *R. sudanica* Chev.. One single species is recognized in this section: *R. sudanica*.

### 3.2. Species Delimitation and Species Complexes

Phylogenetic relationships between species are well to weakly resolved depending on the section, as discussed below.

#### 3.2.1. The Moniliformes (Including Flabellatae) Section

Within the Moniliformes section, species relationships are generally strongly supported (Figure 1 and Figure A1) and several species are recovered as monophyletic (*R. australis*, *R. farinifera*, *R. matombe*) while species limits in others are less clear (*R. textilis*, *R. vinifera*).

Once again, there is a conflict between the concatenated and coalescent analyses. *Raphia textilis* is split into two well-supported clades in the former analysis (Figure A1), while it is recovered as monophyletic with strong support in the latter (Figure 1). Nevertheless, there is little doubt that these samples represent the same species as they are morphologically similar. This is also confirmed by the species delimitation analysis at both levels of α (Figure 1 and Figure A2).

Another result recovered is the close relationship of the two montane species of *Raphia*: *R. ruwenzorica* been included within *R. vinifera*. Both species occupy a similar ecological and altitudinal ranges, despite being geographically separated by ca. 2500 km. *Raphia vinifera*, of which most samples were in the past identified as *R. mambillensis* and now a synonym of *R. vinifera*, occurs mainly in Cameroon and Nigeria, where it grows between 1200 and 2000 m in grassland or open vegetation and is very abundant along streams and rivers [10,32]. *Raphia ruwenzorica* occurs between 800 and 1500 m in the Albertine rift region in eastern Democratic Republic of the Congo and Burundi and has been suggested to grow in “savanna country” along valleys [20,21,33,34]. In addition, both species present similar partial inflorescences that are flat and racket shaped. However, both species differ markedly in their port with *R. ruwenzorica* reported to have a distinct tall stipe reaching up to 15 m [10,21,33], whereas as *R. vinifera* is acaulescent or with a short stipe (less than 1 m; [32]). This, in addition to the 2500+ km separating these species, suggests that they could be recognized as distinct, despite our results. Interestingly, a within species CVL/Albertine rift disjunction has been documented in different taxa such as *Isolona congolana* (Annonaceae [35]) and *Prunus africana* [36].

*Raphia farinifera* is the most widespread species of *Raphia*, occurring from West to East Africa and Madagascar, and has also been reported from the Republic of Congo and Angola [23,34,37,38,39,40,41,42]. Our limited (3) but widespread sampling (West, Central Africa, and Madagascar) of individuals clustered together with maximum support (Figure 1 and Figure A1). However, our species delimitation analysis suggests that the Malagasy individual (R41_T15) is a different species (Figure 1). Interestingly, *Raphia* individuals from Madagascar were initially described as a different species (*R. ruffia* and also once described as *R. tamatavensis* Sadeb.) [22,34] and the name subsequently synonymized with *R. farinifera* [34,40]. In Madagascar, *Raphia* is widely used (one of the most useful palms) and, today at least, not found in natural forests across the island [43]. This has led to the hypothesis that *Raphia* was introduced 1500 years ago during the first wave of human colonization of the island [43]. However, several authors suggested that this species used to occur naturally and abundantly in some places of the Island [22,44]. Our sampling is not extensive enough to answer this question conclusively, but our results suggest that Malagasy individuals might indeed belong to a different species (*R. ruffia*) as concluded by Beccari [22] (p. 53) thus not supporting the recent introduction hypothesis. Finally, *R. farinifera* is recovered as sister to *R. australis*, a relationship already suggested based on morphology [34].

#### 3.2.2. The Raphiate Section

This is one of the most complex and least understood sections. Some of these species are poorly known and rarely collected, sometimes only known from a single poor quality specimen (*R. africana*; *R. longiflora*; *R. mannii*). In our study, we were not able to sample *R. mannii* and *R. longiflora*, thus our results for this section are still incomplete. Overall, the relationships between species in the Raphiate section are weakly to moderately supported in both analyses (Figure 1 and Figure A1). Nevertheless, we do recover monophyletic groups in some species consistent with prior morphological identifications. This is the case for individuals of *R. laurentii* and *R. monbuttorum*, which despite low support are monophyletic. Furthermore, both these species are recovered as sister, with moderate to high support. However, our species delimitation analysis suggests that individuals identified under both species are conspecific (Figure 1A). Indeed, these two species are morphologically similar [19], having clustering stipes covered with straight fibers in addition to having semi-erect inflorescences when young (Figure 1 I), an unusual character within the genus. Nevertheless, it is hard morphologically to consider these two species as conspecific. Indeed, the shape of the rachillae is quite different between these species (Figure 1J,K). *Raphia laurentii* is characterized by rather thick rachillae covered by numerous tightly packed rachis bracts leading to an overall digitate aspect of the rachillae (Figure 1J). In *R. monbuttorum*, the rachillae are thinner and the rachis’ bracts are less tightly packed around the rachillae (Figure 1K). These differences appear to be consistent and provide useful identification characters [19].

*Raphia zamiana* was recently described [7]. Our broad sampling of this species, however, recovers *R. zamiana* as polyphyletic, with individuals grouping into two main clades, flagged as two different species by our species delimitation analysis (Figure 1 and Figure A2). Interestingly, these two species are geographically distinct, with one clade sampled across Gabon and one across Cameroon (Figure 3b,d), the latter containing the type of *R. zamiana*. The Gabon cluster is particularly well supported in both analyses. At this point, however, it is hard to pin point clear morphological characters differentiating these two clusters, as extensive field observations have yet to distinguish them properly.

We sampled two individuals of the Neotropical species *R. taedigera*, both from Brazil. As expected from the morphology of the partial inflorescence [21], this species grouped within the Raphiate section (Figure 1). Both individuals clustered together with strong support, and, in turn, were recovered as sister to either *R. africana* (Figure 1) or *R. palma-pinus* (Figure A1, in both cases with weak support values). Otedoh [21], following certain authors [23,45], suggested that *R. taedigera* was very close morphologically to a species he called *R. vinifera*. However, early on, the taxonomic concept of *R. vinifera* has been confusing, erroneously mistaking this species for a Raphiate type species [23,45]. Mogue Kamga et al. [32] clarified the situation showing that the name *R. vinifera* refers to a Flabelattae species mainly occurring in the CVL. To date, it remains unclear to what species Otedoh and others [21,23,45] were referring to when invoking *R. vinifera*.

Despite these taxonomic confusions, our results provide some indication as to the origin of *R. taedigera* in the Neotropcis. It has been hypothesized that this species originated as a result of vicariance during the breakup of Gondwana [4]. The deeply nested position of *R. taedigera* within the genus does not support this hypothesis. Instead, our results lend some support to the conclusion of Otedoh [10] who suggested that *R. taedigera* did not show any “primitive“ characters within the genus. Otedoh went further to suggest that *R. taedigera* was the result of a recent introduction in South and Central America during the slave trade some 400 years ago [10,11]. Our species delimitation results suggest, however, that *R. taedigera* is a valid species (Figure 1), at least based on the individuals sampled from Brazil. Finally, Otedoh also suggested the presence of *R. taedigera* in coastal west-central Africa [10,11]. However, to date, we have not been able to locate this species in African collections and this hypothesis remains doubtful [1]. Our phylogenetic analyses suggest that *R. taedigera* is genetically quite different from other *Raphia* species (Figure A2) supporting the hypothesis that it must have dispersed to the Neotropics more than 400 years ago. This would fit with paleoecological data from Nicaragua documenting *R. taedigera* pollen over the last 2500 years [9]. A more detailed sampling of *R. taedigera* from the Neotropics together with a dated molecular phylogeny approach will provide a better understanding of the biogeographic history of this interesting trans-Atlantic disjunction.

#### 3.2.3. The Temulentae Section

This section contains the species referred to as the “wine” palms [20,21] with three species previously included in this section (*R. hookeri*, *R. rostrata*, *R. sese*), all of which are sampled here. In addition, our results show that the newly described species, *R. gabonica* [7], is also part of the Temulentae section. This was not clear at the time of the publication as the partial inflorescence suggested a possible relationship with the Moniliformes section [7]. Overall, species identified based on morphology clustered together (e.g., *R. gabonica*, *R. sese*) with strong or low support (Figure 1 and Figure A2). Nevertheless, all four species show a very close phylogenetic proximity, suggesting that this section could be regarded as a species complex. Indeed, depending on the level of stringency used for the SODA analysis, our species delimitation analysis recovered either seven distinct species or one single species (Figure 1 and Figure A2). It is important to note that changing levels of α did not impact species delimitation in the other sections. Morphologically, however, these species are different and can easily be identified in the field, which is partly supported by our phylogenetic analysis. For example, *R. gabonica* resembles *R. hookeri* in the clearly visible single stipe covered with characteristic curly fibers, but differs markedly by being a *terra firme* low-density species with thin (Moniliformes-like) and densely packed rachillae. In contrast, *R. hookeri* is a swampy species, growing in large, monodominant stands with robust and more evenly-spaced rachillae [7,19]. In the same way, *R. rostrata* is characterized by a small but clustering stipe with mixed curly and straight hanging fibers and occurs along rivers with strong currents [19].

*Raphia hookeri* is recovered here as polyphyletic, possibly including four different cryptic species. This is one of the most important, abundant, and widespread *Raphia* species and its overall morphology is rather constant across its range. However, individuals appear to be geographically structured like in *R. zamiana* (see above). Interestingly, this mirrors patterns of genetic structure recovered across a wide range of central African plant species [46,47], including *R. zamiana*.

## 4. Materials and Methods

### 4.1. Species Sampling, Library Preparation, and DNA Sequencing

We sampled a total of 56 individuals (see Table A1 in Appendix A for details) representing 18 out of the 21 species accepted to date [7,19] and representing all sections described by Otedoh [21]. In order to collect proper material for sequencing, several field trips were undertaken across several African countries including Ivory Coast, Ghana, Gabon, Cameroon, Angola, and the Demographic Republic of the Congo between 2012 and 2017. We were not able to access material from three accepted species: *R. gentiliana*, *R. mannii*, and *R. longiflora*. We sampled two to seven individuals per species in order to test for monophyly. However, only a single specimen was available for *R. ruwenzorica*. Finally, we sampled four species within Calamoideae as outgroups: *Eremospatha cabrae*, *Eremospatha quinquecostulata*, *Laccosperma cristalensis*, and *Mauritiella armata* following [25,48]. We extracted DNA from leaves dried in silicagel, except for one individual of *R. taedigera* and the only individual of *R. ruwenzorica* for which DNA was extracted from herbarium material.

Methods for DNA extraction, preparation of sequencing libraries, hybridization, Illumina MiSeq DNA sequencing, and read cleaning followed [19]. In brief, barcoded Illumina libraries were constructed based on a modified protocol of Rohland and Reich [49]. We hybridized DNA to defined exons using the palm-specific nuclear baiting kit of Heyduk et al. [26]. This kit allows for sequencing exons from 176 nuclear genes across the palm family.

### 4.2. Contig Assembly and Multi-Sequence Alignment

We used HybPiper (v1.2) [50] to process our cleaned reads (following [19]) to obtain sequences corresponding to the target exons plus associated intronic sequence data (referred to as supercontigs).

Briefly, we demultiplexed the data and removed adapters. Reads were filtered according to their length (>35 bp) and quality mean values (Q > 30). We trimmed 6 bp of sequences to ensure removal of barcodes when sequences were shorter than 150 bp. Reads were then mapped to target exons and successfully mapped reads were assembled into contigs. These contigs were then aligned to their associated target exon sequence. If contigs were slightly overlapping [50], they were combined into “supercontigs” which contain both target and off-target sequence data. We aligned each set of supercontigs using MAFFT (v7.305) [51] with the –auto option and cleaned these alignments with GBLOCKS (v0.91b) [52] using the default parameters and all allowed gap positions.

To identify a suitable set of loci for phylogenetic inference, we selected only those supercontigs that had 75% of their exon length reconstructed in at least 25% of individuals (referred to as 75/25). We used only those loci in which at least 75% of the exon length was recovered because the use of fragmented sequences is known to increase gene tree error, whereas the number of individuals has little effect as long as the gene tree is accurate [53].

#### Paralog Identification

HybPiper flags potential paralogs when multiple contigs are discovered mapping well to a single reference sequence. We ran hybpiper on the 837 exons that made up the baiting kit [26], identified flagged loci, and constructed exon trees using RAxML (v8.2.9) [54]. We examined each tree to determine whether putative paralogs formed a species clade. When sequences concerning more than three individuals were flagged for a locus, we examined whether the ’main’ and alternative sequences formed separate clades. If so, this locus was classified as a paralog and discarded from the dataset. For each gene, we then calculated at the proportion of exons that we confirmed as paralogs after inspection. If this proportion was <50%, we removed the entire gene from our analyses.

### 4.3. Coalescent Phylogenetic Inference

Individual gene trees were constructed with 100 bootstraps and the GTRGAMMA model using RAxML (v8.2.9) [54] (option “-f a“). If, after inference, branches had bootsrap support values >10, they were collapsed using the program nw_ed [55] because this approach has been shown to improve the accuracy of ASTRAL [31]. We used the selected 75/25 gene trees as our input to run ASTRAL-III (v5.5.11) [31] using the default options.

### 4.4. Species Delimitation

After constructing our ASTRAL tree, we used the associated approach SODA [56]. Simulations using this approach have shown it to be of similar accuracy or more accurate [56] than other popular species delimitation methods such as BPP [57] at a fraction of the computational cost. SODA uses frequencies of quartet topologies to determine if each branch in a guide tree inferred from gene trees (i.e., the ASTRAL tree from above) is likely to have a positive length. This identifies where in the tree coalescence is random, and where it is non-random. It then uses the results to infer a new, extended species tree that defines boundaries among species. We used two cut-off values of α (confidence level): 0.01 and 0.005.

### 4.5. Maximum-Likelihood Phylogenetic Inference

After suitable loci were identified, we filled any missing individuals in each alignment with an empty sequence. We then concatenated all aligned loci using the *pxcat* function in the program phyx [58]. We used IQ-TREE (v1.6.8; [59]) to infer a maximum likelihood tree of all individuals. We partitioned our dataset so that each supercontig had a separate substitution model and used the following options when running the program: “-m MFP + MERGE -rcluster 10 -bb 1000 -alrt 1000”. We selected the optimal partitioning scheme using ModelFinder [60], choosing the best model based on Bayesian Information Criterion (BIC) score and merging genes until model fit stopped increasing. We also used rcluster [61] to decrease computational load. We made use of the ultrafast bootstrapping ([62]; 1000 replicates) and the SH-like approximate likelihood ratio test ([63]; 1000 replicates) to assess branch support in the tree.

### 4.6. SNP Calling

To call SNPs, we first used SeCaPr (v1.1.4; [64]) to build a psuedoreference. After filtering out low coverage and paralogous loci, consensus sequences are built and combined to form a reference file that is closer to the study group than the original, and will recover more data. We mapped our cleaned, paired reads to this new, dataset-specific reference using BWA (v0.7.12; [65]). Duplicates were removed and we called SNPs using the program HaplotypeCaller in GATK (v4.0; [66]). We applied thresholds to mapping quality (>40%) depth (>25), quality by depth (>2), minimum quality across all individuals (>10) and minor allele frequency (>0.01) to filter SNPs using bcftools (v1.8; [67]). We kept only biallelic SNPs and excluded monomorphic sites.

### 4.7. Genetic Clustering

We performed Discriminant Analysis of Principal Components (DAPC) [68] to identify genetic clusters in two species complexes of *Raphia*. We used the function *find.clusters* in the R package ‘adegenet’ [69] to infer the number of clusters using successive K-means with 100,000 iterations per value of k up to k = 20. We used BIC to identify the best-fitting number of clusters. We then used the function *dapc* [68] to define the diversity among the clusters identified. We chose the optimum number of axes to use with the function *optim.a.score*.

## 5. Conclusions

Our results provide a new step forward in understanding the phylogenetic relationships and taxonomy within this major African palm genus. We show that the morphological sections based on partial inflorescence shape defined by Otedoh [21] are relatively robust overall. Sections Obclavatae, Temulentae, and Raphiate are recovered as monophyletic with good support, while sections Moniliformes and Flabellatae are not. We thus redefined these later two sections into sections Erectae, composed of *R. regalis*, and Moniliformes, including all species previously included in Flabellatae. Our results also uncover important species delimitation problems defined here as species complexes (*R. hookeri, R. zamiana*) that must be solved if we are to have a thorough understanding of *Raphia* systematics. Given the economic and ecological importance of *R. hookeri*, clarifying its species delimitation will be important in the future. Different approaches could rely on more in-depth population level studies using more variable markers (e.g., microsatellites) combined with detailed morphometric measurements as has been done in other African tree species, e.g., [70,71,72]. A better comprehension of the taxonomy and the phylogenetic relationships in *Raphia* represents a fundamental tool towards the proposal of conservation strategies aiming to characterize genetic and morphologic diversity in this ecological and economically important genus. Finally, we show here that the Heyduk et al. baiting kit [26] is useful for understanding relationships within the *Raphia* genus and between species as was shown in other groups e.g., [73], although it appears to be limited for untangling species complexes. Resolving relationships within *Raphia* will thus rely on more data, including increased infra-species sampling, detailed morphological studies in certain species, and larger baiting kits, e.g., [74].

## Figures and Tables

**Figure 1 plants-09-00549-f001:**
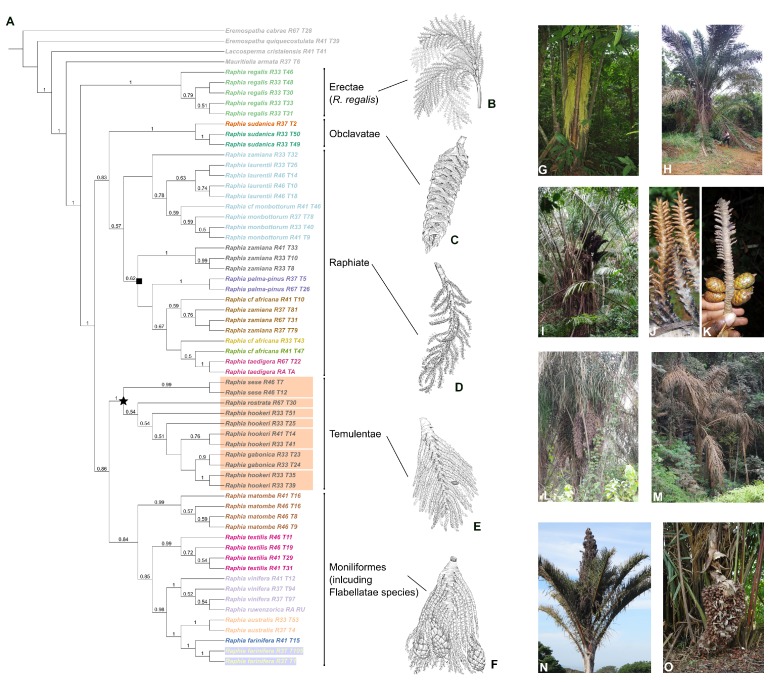
(**A**) Cladogram of the genus *Raphia* inferred using 85 gene trees and ASTRAL. Values of local posterior probabilities (LPP) equal or above 0.5 are shown above the branches. Branch lengths are represented in Figure A2. Individuals are color coded based on the hypothesis of species delimitation inferred using SODA with α = 0.01. For a single clade, the Temulentae section marked with a black star and referred to as the “hookeri“ complex in the main text, varied between the two values of α used here. The orange boxes represent the species limits using SODA with a more stringent value of α = 0.005. The black square shows the “zamiana“ complex clade. Tip names contain the species name as well as the sequencing ID. (**B**) *R. regalis* partial inflorescence representing the Erectae section (described here, see discussion). (**C**) *R. sudanica* inflorescence, representing the Obclavatae section. (**D**) *R. palma-pinus* inflorescence, representing the Raphiate section. (**E**) *R. hookeri* inflorescence, representing the Temulentae section. (**F**) *R. farinifera* inflorescence, representing the Moniliformes (and ex Flabellatae) section. (**G**) *R. regalis*, note the acaulescent habitat and inflorescences subtended by the leaves (Couvreur 398, Cameroon). (**H**) *R. zamiana* (Mogue Kamga 17, Gabon). (**I**) *R. monbuttorum* (Couvreur 1212, Cameroon), notice the semi-erect inflorescences. (**J**) detail of *R. monbuttorum* rachillae (Couvreur 1212, Cameroon). (**K**) detail of *R. laurentii* rachillae (Mogue Kamga 39, Democratic Republic of Congo). (**L**) *R. hookeri* (no voucher, Cameroon). (**M**) *R. gabonica* (Mogue Kamga 22, Gabon). (**N**) *R. australis* (no voucher, South Africa, Kirstenbosch Botanic Garden). (**O**) Inflorescence of *R. vinifera* (Couvreur 638, Cameroon). (**B**–**F**) line drawings by Mary Grierson reproduced with permission from [23] and Royal Botanic Gardens, Kew (U.K.); Photos (**G**–**J**), (**L**–**O**) T.L.P. Couvreur; Photo (**K**) S. Mogue Kamga.

**Figure 2 plants-09-00549-f002:**
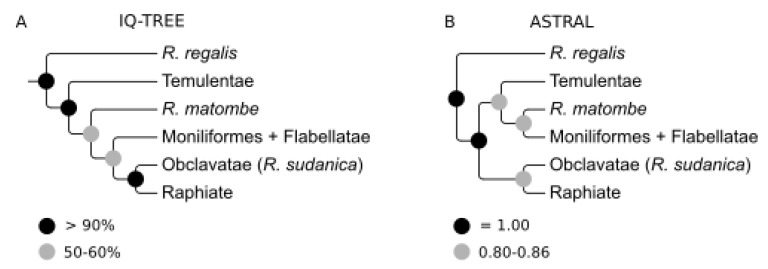
Major incongruences between the (**A**) concatenation (IQ-TREE) and (**B**) gene tree (ASTRAL) phylogenetic approaches. Both trees have been modified to show the relationships among major *Raphia* clades. Support values are indicated on the nodes as either (a) bootstrap or (b) local posterior probability.

**Figure 3 plants-09-00549-f003:**
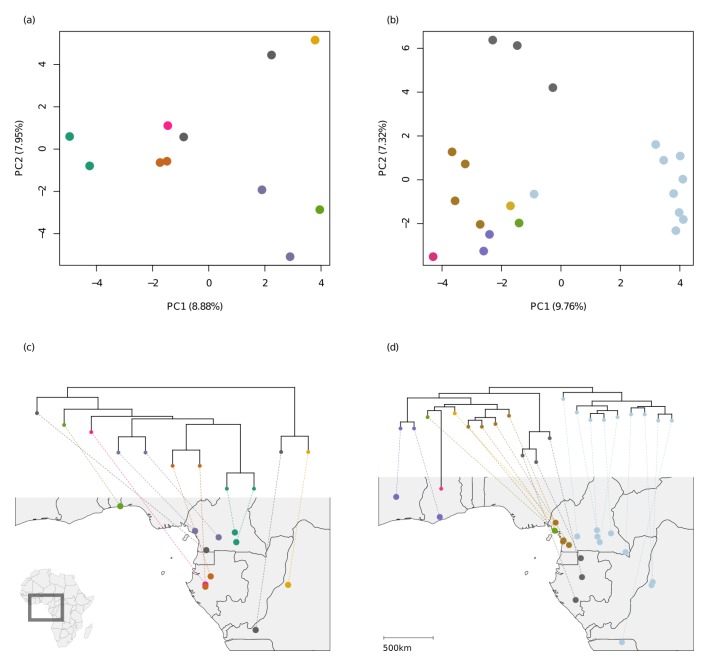
(**a**) scatterplot of *R. hookeri* complex based on 915 SNPs; (**b**) scatterplot of *R. zamiana* complex based on 1627 SNPs. Clades representing the (**c**) *R. hookeri* and (**d**) *R. zamiana* complexes were extracted from the ASTRAL (Figure 1) tree and linked to their locations on a map of central Africa (area in context on inset map). Individuals are colored by the colors corresponding to SODA species delimitation, an approach that uses gene tree topologies to determine whether coalescence is random or non-random and delimit species based on this. We depict SODA results for two different values of α (confidence): 0.01 in (**a**,**c**) and 0.005 in (**b**,**d**). An individual belonging to *R. taedigera* (RA_TA) is not shown in panels (**a**) and (**c**) due to missing data.

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
