# Peer review of "Unraveling the Phylogenomic Relationships of the Most Diverse African Palm Genus *Raphia* (Calamoideae, Arecaceae)"

_plants, 2020, doi:10.3390/plants9040549_

Round 1
Reviewer 1 Report
This work presents a phylogenomic framework of the genus Raphia, the most diverse genus of African palms. Although it is a genre with a particular relevance also from an economic point of view, to date there were still no intrageneric phylogenetic studies aimed at putting an order in its taxonomy. I want to clarify that I am not a specialist in the field, but in general, it is a valid article. The introduction is well written and poses the basis for the hypotheses tested
The article as a whole is written in a clear, concise and orderly manner and is ready to read even for readers not necessarily specialized in phylogenomics. According to the authors' conclusions, the morphology of the inflorescence supports the proposed infrageneric classification. The fate that the authors highlight the aspects not fully clarified by their analysis and the following interpretative limitations is also commendable.
Figure 3 is interesting in that it tries to relate the results of the ASTRAL tree analysis to the geographical distribution of the species. The caption could be improved by referring more explicitly to the SODA species delimitation. As a general comment, the variance explained by the two components is rather low (one could comment more on the possible reason).
Author Response
Reviewer 1
This work presents a phylogenomic framework of the genus Raphia, the most diverse genus of African palms. Although it is a genre with a particular relevance also from an economic point of view, to date there were still no intrageneric phylogenetic studies aimed at putting an order in its taxonomy. I want to clarify that I am not a specialist in the field, but in general, it is a valid article.
We thank the reviewer for their useful comments.
The introduction is well written and poses the basis for the hypotheses tested
The article as a whole is written in a clear, concise and orderly manner and is ready to read even for readers not necessarily specialized in phylogenomics. According to the authors' conclusions, the morphology of the inflorescence supports the proposed infrageneric classification. The fate that the authors highlight the aspects not fully clarified by their analysis and the following interpretative limitations is also commendable.
Figure 3 is interesting in that it tries to relate the results of the ASTRAL tree analysis to the geographical distribution of the species. The caption could be improved by referring more explicitly to the SODA species delimitation.
We have modified the caption to more explicitly. explain the SODA approach and which results we present (legend of Fig. 3).
Individuals are coloured by the colours corresponding to SODA species delimitation, an approach that uses gene tree topologies to determine whether coalescence is random or non-random and delimit species based on this. We depict SODA results for two different values for alpha (confidence): 0.01 in (a) & (c) and 0.005 in (b) & (d).
As a general comment, the variance explained by the two components is rather low (one could comment more on the possible reason).
This is typical from using SNP data, due to the complexity of genomes and their evolution. It is difficult to say from a PCA the exact mechanism at play, but we have added a passage pointing this out (L164).
The first two PCs in both analyses explained 7-10% of the variance in the dataset. This is a still less than a quarter of the total variance in each case, but this may be expected from our relatively complex dataset of almost 100 genes with independent histories.
Reviewer 2 Report
Title: UNRAVELLING THE PHYLOGENOMIC RELATIONSHIPS OF THE MOST DIVERSE AFRICAN PALM GENUS Raphia (CALAMOIDEAE, ARECACEAE)
Nice and interesting study, I appreciate the effort of the authors. All the methods that were used are actual and relevant for the aim of the study. I suggest to write a methods section a little detailed, especially the bioinformatic section. The results and discussion are written well, no changes are needed. My only minor recommendation is to provide information about existing molecular genetic data for the genus Raphia in introduction.
Author Response
Nice and interesting study, I appreciate the effort of the authors. All the methods that were used are actual and relevant for the aim of the study.
We thank the reviewer for their helpful comments.
I suggest to write a methods section a little detailed, especially the bioinformatic section.
We have added additional details to the methods (L415)
Briefly, we demultiplexed the data and removed adapters. Reads were filtered according to their length (>35 bp) and quality mean values (Q > 30). We trimmed 6 bp of sequences to ensure removal of barcodes when sequences were shorter than 150 bp. Reads were then mapped to target exons and successfully mapped reads were assembled into contigs. These contigs were then aligned to their associated target exon sequence. If contigs were slightly overlapping they were combined into “supercontigs” which contain both target and off-target sequence data. We aligned each set of supercontigs using MAFFT (v7.305) with the --auto option and cleaned these alignments with GBLOCKS (v0.91b) using the default parameters and all allowed gap positions.
The results and discussion are written well, no changes are needed.
My only minor recommendation is to provide information about existing molecular genetic data for the genus Raphia in introduction.
We already had a section detailing the existing molecular genetic data and previous phylogenetic hypotheses in Raphia (L91 - 94). There were very few analyses prior to our study. We added extra info about what markers were used.